elLIFE

# Rapid decline of bacterial drug-resistance in an antibiotic-free environment through phenotypic reversion

Anett Dunai[1,2†], Réka Spohn[1†], Zoltán Farkas[1†], Viktória Lázár[1‡], Ádám Györkei[1], Gábor Apjok[1,2], Gábor Boross[1§], Balázs Szappanos[1], Gábor Grézal[1], Anikó Faragó[2,3], László Bodai[3], Balázs Papp[1], Csaba Pál[1*]

[1]Synthetic and Systems Biology Unit, Institute of Biochemistry, Biological Research Centre, Hungarian Academy of Sciences, Szeged, Hungary; [2]Doctoral School in Biology, Faculty of Science and Informatics, University of Szeged, Szeged, Hungary; [3]Department of Biochemistry and Molecular Biology, University of Szeged, Szeged, Hungary

**Abstract** Antibiotic resistance typically induces a fitness cost that shapes the fate of antibiotic-resistant bacterial populations. However, the cost of resistance can be mitigated by compensatory mutations elsewhere in the genome, and therefore the loss of resistance may proceed too slowly to be of practical importance. We present our study on the efficacy and phenotypic impact of compensatory evolution in *Escherichia coli* strains carrying multiple resistance mutations. We have demonstrated that drug-resistance frequently declines within 480 generations during exposure to an antibiotic-free environment. The extent of resistance loss was found to be generally antibiotic-specific, driven by mutations that reduce both resistance level and fitness costs of antibiotic-resistance mutations. We conclude that phenotypic reversion to the antibiotic-sensitive state can be mediated by the acquisition of additional mutations, while maintaining the original resistance mutations. Our study indicates that restricting antimicrobial usage could be a useful policy, but for certain antibiotics only.
DOI: https://doi.org/10.7554/eLife.47088.001

*For correspondence:
cpal@brc.hu

†These authors contributed equally to this work

Present address: ‡Faculty of Biology, Technion – Israel Institute of Technology, Haifa, Israel; §Department of Biology, Stanford University, Stanford, United States

Competing interests: The authors declare that no competing interests exist.

## Introduction

Steady antibiotic overuse has led to the rise and spread of multidrug-resistant bacteria, and can potentially reduce the number of therapeutic options against several dangerous human pathogens. Resistance seriously impacts the effectiveness of treatment, and increases the risk of complications and fatal outcome (*Mensah Abrampah et al., 2018*). Strict policies that aim to restrict antimicrobial usage in clinical settings may offer a solution to this problem (*Guay, 2008*). Such strategies implicitly presume that resistance leads to reduced bacterial fitness in an antibiotic-free environment, and therefore these resistant populations should be rapidly outcompeted by antibiotic-sensitive variants. In theory, the extent of fitness costs determines the long-term stability of resistance, and consequently, the rate by which the frequency of resistant bacteria decreases in an antibiotic-free environment. Resistance mutations frequently incur fitness costs in the laboratory (*Andersson and Levin, 1999*; *Levin et al., 1997*), as such mutations impair essential cellular processes, such as transcription, translation, or cell-wall biogenesis. There is a negative correlation between measured fitness costs and prevalence in clinical settings (*Basra et al., 2018*; *Praski Alzrigat et al., 2017*; *Trindade et al., 2009*). This would suggest that fitness costs shape the propagation of antibiotic resistant bacteria in the clinics. However, in other cases, such deleterious side effects of resistance mutations are

undetectable, and resistance can even confer benefits in specific, antibiotic-free environmental settings (*Maharjan and Ferenci, 2017*).

It is frequently assumed that such compensatory mutations mitigate the fitness costs of resistance mutations without affecting the level of resistance. As the range of targets for compensation is much broader, compensatory mutations are more likely than the reversion of resistance mutations. If compensatory mutations are indeed widespread, pathogens can reach both high level of resistance and high fitness. For these reasons, reversion to the original antibiotic-sensitive state under prudent antibiotic use may proceed so slowly that it will have no practical importance in the clinic (*Andersson, 2006*; *Andersson and Hughes, 2010*; *Nicoloff et al., 2019*; *Schrag et al., 1997*).

Several prior laboratory studies support the argument above. It has been reported that antibiotic resistance is stably maintained as a result of compensatory mutations (*Björkman et al., 1999*; *Johanson et al., 1996*; *Marcusson et al., 2009*), however, instances of compensatory mutations have not been generalized to clinical settings. Several published clinical studies indicate that limited antibiotic use can cause rapid changes in the frequency of resistance. In certain cases, resistance tends to decline both in individual patients and at the community level following restricted antibiotic use (*Butler et al., 2007*; *Dagan et al., 2008*; *Gottesman et al., 2009*; *Kristinsson, 1997*). For example, the frequency of erythromycin-resistant *Streptococcus pyogenes* steadily declined as a result of reduced use of macrolides in Finnish hospitals (*Seppälä et al., 1997*). By contrast, months of reduction of sulfamethoxazole clinical treatment in the United Kingdom failed to reach reduced resistance levels (*Enne et al., 2001*).

This disagreement between clinical observations and laboratory studies could have multiple reasons. First, antibiotic treatments frequently fail to completely eradicate antibiotic-sensitive bacteria from the population. Following treatment, antibiotic sensitive bacteria with high fitness could rapidly spread in the population, leading to rapid loss of resistance. Second, compensatory evolution may be limited in nature and in clinical settings (*Brandis et al., 2012*; *Hall and MacLean, 2011*; *Vogwill and MacLean, 2015*). Indeed, most laboratory studies focused on resistance to a single drug (*Brandis et al., 2012*; *Maisnier-Patin et al., 2002*; *Qi et al., 2016*), thus compensatory evolution in multidrug-resistant bacteria has remained largely unexplored (*Vogwill and MacLean, 2015*; but see *Moura de Sousa et al., 2017*). This latter shortcoming is especially noteworthy, because several multidrug-resistant bacteria are difficult to treat with current antibiotic treatment, and therefore the issue of compensatory evolution is especially relevant. Indeed, there is an urgent need to understand the mechanisms that mitigate the costs of multiple – potentially epistatically interacting – mutations in multidrug-resistant pathogens (*Trindade et al., 2009*; *Wong, 2017*), which clearly cannot be achieved in studies focusing on single mutants only.

In this work, we have studied 23 drug-resistant *E. coli* strains that carry 2 to 13 mutations (*Lázár et al., 2014*). We have found that 60 days of laboratory evolution under antibiotic-free conditions has led to a rapid decline of resistance to certain, but not all antibiotics. This decline in resistance has not resulted from strict reversion mutations that recapitulate wild-type bacteria at the molecular level. Rather, the mutated genes are functionally related, but not identical to those conferring antibiotic resistance. Notably, these mutations perturb membrane-permeability and the activity of regulons involved in defense against drugs and related stresses. They partially restore wild-type fitness in the antibiotic-free medium, and also reduce the level of antibiotic-resistance, at least against certain antibiotics. These considerations could help to identify antibiotics where restricting antimicrobial usage could be a useful policy.

## Results

### Evolution of resistance and fitness cost

In a previous work we had performed adaptive laboratory evolution under gradually increasing antibiotic dosages (*Lázár et al., 2014*). Parallel evolving populations *of E. coli* K-12 BW25113 were exposed to 1 of 12 antibiotics (*Table 1*). Adapting populations were found to display an up to 328-fold increment in their minimum inhibitory concentrations (MICs) relative to the wild-type strain. To elucidate the underlying molecular mechanisms of resistance, 60 antibiotic-resistant strains (one clone per each population) were previously isolated and subjected to whole genome sequencing

**Table 1.** Antibiotics employed and their modes of actions.

Functional classification (molecular target) is based on previous studies (*Girgis et al., 2009*; *Yeh et al., 2006*). These antibiotics are widely used in clinical practice, are well-characterized and cover a wide range of modes of actions. For three-letter codes (abbreviations), the AAC's standard (https://aac.asm.org/content/abbreviations-and-conventions) was used.

| Antibiotic name | Abbreviation | Antibiotic class | Molecular target | Effect | Number of T0 lines displaying significant fitness cost |
|---|---|---|---|---|---|
| Ampicillin | AMP | Penicillin | Cell wall | Bactericidal | 1 |
| Cefoxitin | FOX | Cephalosporins | Cell wall | Bactericidal | 1 |
| Ciprofloxacin | CIP | Quinolones | Gyrase | Bactericidal | 2 |
| Nalidixic acid | NAL | Quinolones | Gyrase | Bactericidal | 0 |
| Nitrofurantoin | NIT | Nitrofurantoin | Multiple mechanisms | Bactericidal | 2 |
| Kanamycin | KAN | Aminoglycosides | Protein synthesis, 30S | Bactericidal | 4 |
| Tobramycin | TOB | Aminoglycosides | Protein synthesis, 30S | Bactericidal | 4 |
| Tetracycline | TET | Tetra-cyclines | Protein synthesis, 30S | Bacteriostatic | 2 |
| Doxycycline | DOX | Tetra-cyclines | Protein synthesis, 30S | Bacteriostatic | 1 |
| Chloramphenicol | CHL | Chloramphenicol | Protein synthesis, 50S | Bacteriostatic | 2 |
| Erythromycin | ERY | Macrolides | Protein synthesis, 50S | Bacteriostatic | 0 |
| Trimethoprim | TMP | Trimethropim | Folic acid biosynthesis | Bacteriostatic | 4 |

DOI: https://doi.org/10.7554/eLife.47088.002

analysis. The resistance mutations identified generally affected the drug targets, efflux pumps, porins and proteins involved in cell envelope biogenesis.

In the current study, we estimated fitness by measuring individual fitness of each of the 60 antibiotic-resistant strains and the corresponding wild-type strain in an antibiotic-free medium (*Figure 1—figure supplement 1*). In total, 38% of the antibiotic-resistant strains (N = 23) showed significantly reduced growth compared to the ancestral wild-type strain (*Figure 1—source data 1* . Fitness cost varied substantially across antibiotics (*Figure 1*). Most notably, laboratory-adapted aminoglycoside-resistant strains displayed especially low fitness in the antibiotic-free medium, but the reasons are unclear. However, we note that akin to clinically observed small-colony variants, the major targets of selection under aminoglycoside stress were the translational machinery and a broad class of genes that shape the electrochemical potential of the outer membrane (*Lázár et al., 2013*; *Vestergaard et al., 2016*).

In agreement with prior studies (*Marcusson et al., 2009*; *Melnyk et al., 2015*; *Merker et al., 2018*), strains with especially low fitness values in the antibiotic-free medium were found to carry more resistance mutations (Spearman's correlation, $\rho$ = –0.7488; p<0.0001; N = 23) and display especially high levels of resistance (Spearman's correlation, $\rho$ = –0.658; p=0.0006; N = 23).

## Laboratory evolution of antibiotic-resistant strains in an antibiotic-free medium

To investigate potential changes in resistance phenotypes upon evolution in an antibiotic-free environment, we initiated parallel laboratory evolutionary experiments with 23 out of the 60 antibiotic-resistant strains (1 to 4 strains per antibiotic, see Materials and methods). All these strains exhibited significant fitness costs in antibiotic-free environment. Six parallel populations per antibiotic-resistant strain were cultivated in a standard, antibiotic free-medium, resulting in 138 independently evolving populations. The populations were propagated for 60 transfers (approximately 480 generations) by diluting 1% of the saturated cultures into fresh medium every 24 hr. From each evolved population, we selected a single, representative clone for further analysis (see Materials and methods). Throughout the paper, T0 and T60 refer to the 23 antibiotic-resistant strains and the corresponding 138 evolved lines from the final day of the antibiotic-free evolutionary experiment, respectively.

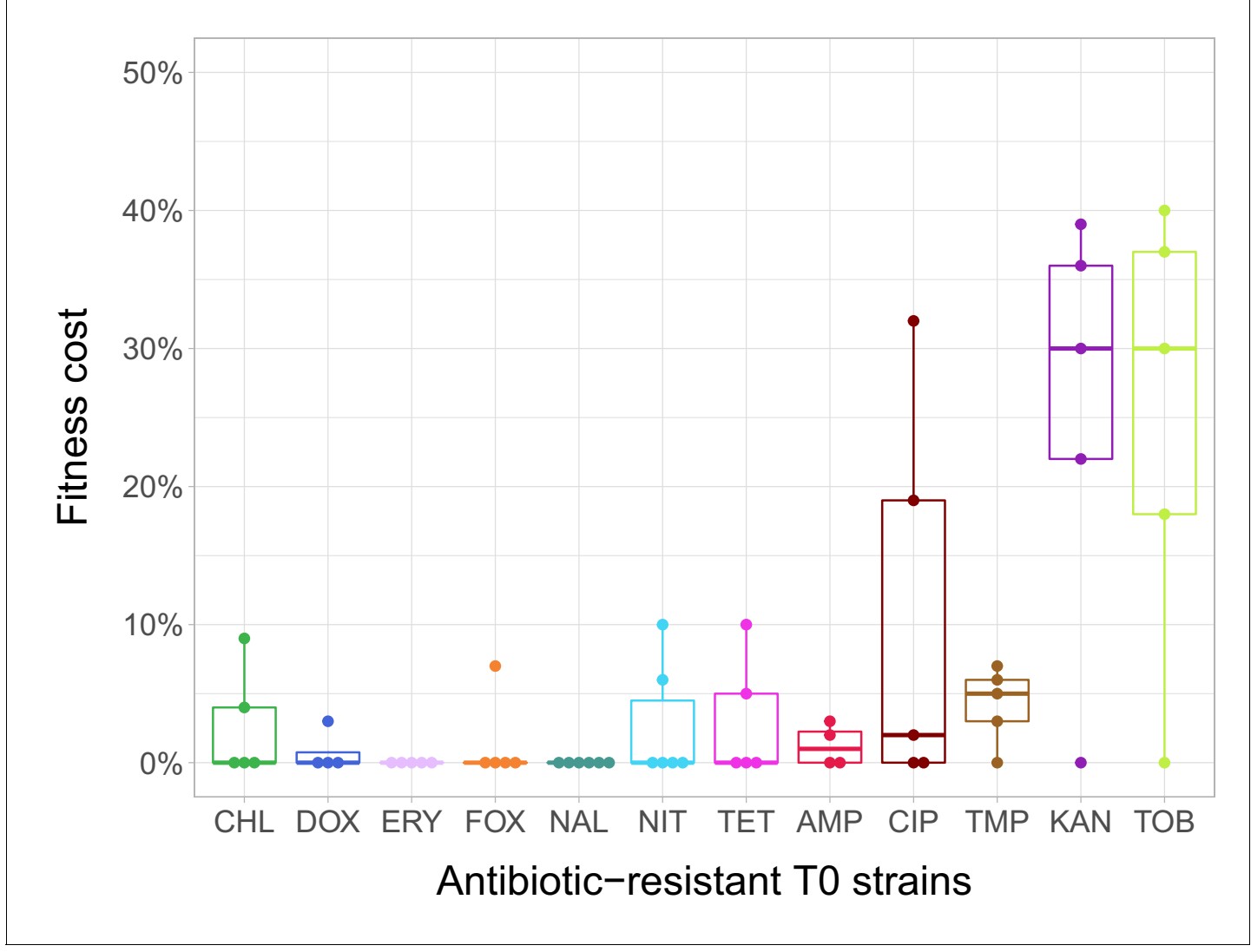

**Figure 1.** Fitness cost of antibiotic-resistant T0 strains. The fitness of each strain was measured as the area under the bacterial growth curve recorded in an antibiotic-free medium. Fitness cost was calculated from the absolute fitness of T0 and wild-type strains using the following equation: $1 - W_{T0}/W_{wild-type}$, where $W_{T0}$ and $W_{wild-type}$ indicate fitness of the T0 and wild-type strains, respectively. Strains adapted to two aminoglycosides – kanamycin (KAN) and tobramycin (TOB) – generally exhibit an especially high fitness cost, while adaptation to erythromycin (ERY) and nalidixic acid (NAL) evoked no measurable fitness cost. For all further analyses only the T0 strains exhibiting a significant fitness cost were used. For antibiotic abbreviations, see *Table 1*. Boxplots show the median, first and third quartiles, with whiskers showing the 5th and 95th percentiles of the fitness cost per antibiotic. Individual data points represent the median of three biological replicates (five technical measurements) for each of the 60 antibiotic-resistant strains (4–6 strain per antibiotic). Source file is available as *Figure 1—source data 1*.

DOI: https://doi.org/10.7554/eLife.47088.003

The following source data and figure supplement are available for figure 1:

**Source data 1.** Initial fitness cost of the T0 antibiotic-resistant strains.
DOI: https://doi.org/10.7554/eLife.47088.005
**Figure supplement 1.** Schematic outline of the evolutionary experiments.
DOI: https://doi.org/10.7554/eLife.47088.004

As in our previous work, the fitness of each of the 23 T0 strains and the 138 T60 lines was measured by estimating the area under the growth curve recorded in the same liquid antibiotic-free medium. 51% of the T60 lines were found to exhibit significantly improved fitness, and some of these lines even approximated wild-type fitness (*Figure 2—source data 1*). The analysis has revealed

major differences in relative fitness across strains adapted to different antibiotics (*Figure 2*), indicating that adaptation is mainly driven by the set of resistance mutations present in the T0 strains.

## Rapid loss of drug resistance in an antibiotic-free medium

Next, we investigated how laboratory evolution in an antibiotic-free medium shapes antibiotic resistance. For this purpose, we first measured the minimum inhibitory concentrations (MIC) in 71 T60 lines showing significant fitness improvement, as well as in the corresponding 20 T0 strains, against a set of 11 antibiotics (see *Table 1* and Materials and methods). Using the CLSI resistance break-

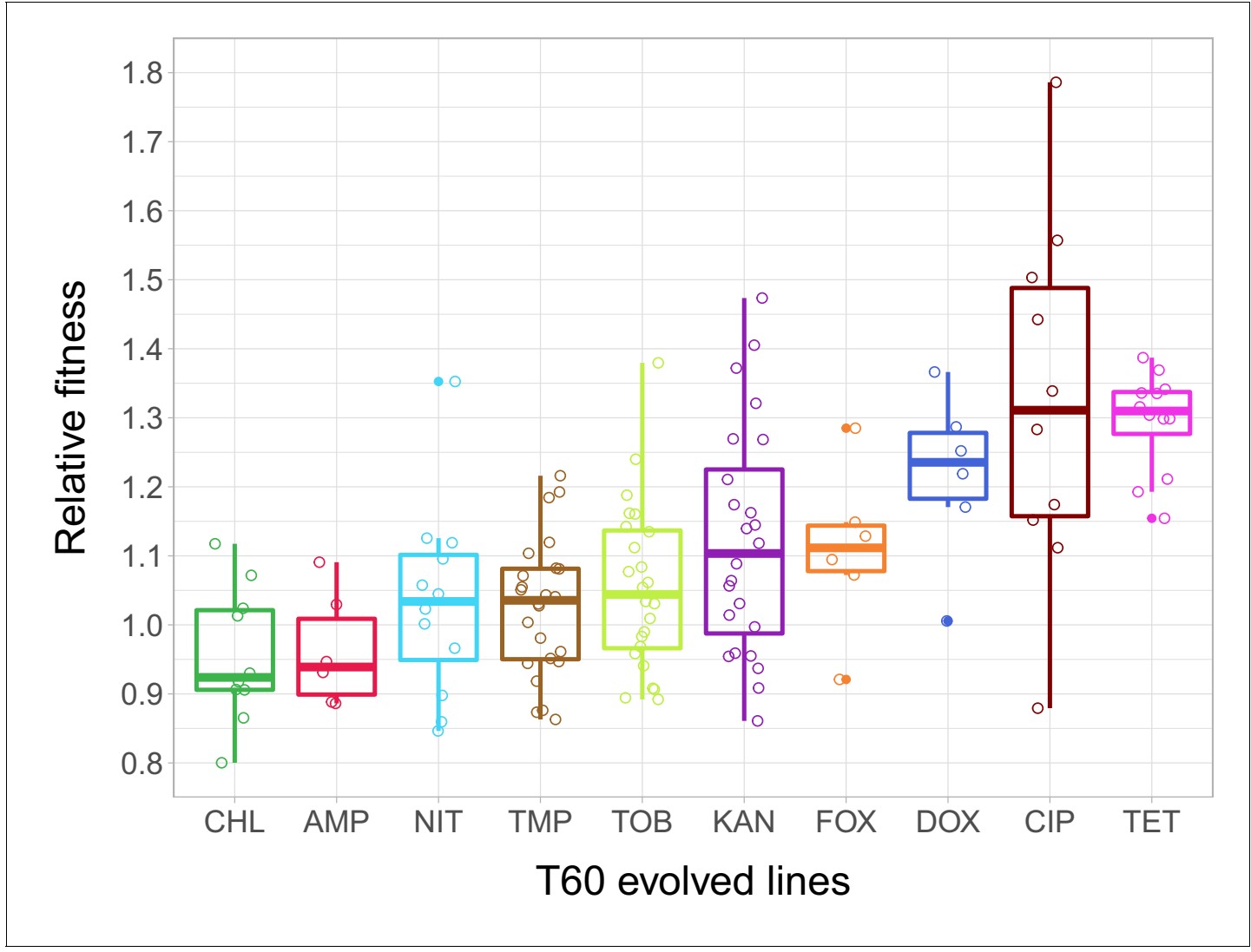

**Figure 2.** Relative fitness of the T60 evolved lines. The figure shows the relative fitness of 6 parallel lines of 23 antibiotic-adapted strains (**T0**) after laboratory evolution in an antibiotic-free medium. Relative fitness was calculated by normalizing the T60 fitness to the fitness of the corresponding T0 antibiotic-resistant strain. Fitness was measured as the area under the growth curve in an antibiotic-free medium. Relative fitness varied significantly across populations initially resistant to different antibiotics (N = 138, p<0.0001, Kruskal–Wallis test). For antibiotic abbreviations, see *Table 1*. Boxplots show the median, first and third quartiles, with whiskers showing the 5th and 95th percentiles of the relative fitness of all T60 evolved lines originally adapted to different antibiotics. Individual data points represent the median relative fitness of each of the 138 T60 evolved lines (3 biological and five technical replicates per each). Source file is available as *Figure 2—source data 1*.

DOI: https://doi.org/10.7554/eLife.47088.006

The following source data is available for figure 2:

**Source data 1.** Fitness improvement reached following 60 day evolution in an antibiotic free environment.

DOI: https://doi.org/10.7554/eLife.47088.007

point cut-offs (CLSI Approved Standard M100, 29th Edition), we categorized each strain as being resistant (R), intermediate (IM) or susceptible (S) to each investigated antibiotic. As expected, the T0 strains generally displayed reduced susceptibility to multiple antibiotics (MIC > R or MIC > IM, see *Figure 3—source data 1*). Based on this categorization, we next examined whether resistance to these antibiotics was maintained or reduced during the laboratory evolution in the antibiotic-free medium. For this purpose, we compared the resistance levels of the 71 T60 lines that displayed significant fitness improvement in the antibiotic-free environment to that of the corresponding T0 strains. We focused on antibiotics to which the corresponding T0 strain exhibited resistance, leading to a total of 195 antibiotic-T60 line combinations.

We have found that resistance declined in as high as 54.8% of the antibiotic-T60 line combinations following evolution under antibiotic stress-free conditions (*Figure 3A and B*, *Figure 3—source data 1*). However, the extent of resistance decline depended on the antibiotic considered. For example, doxycycline and tetracycline resistance was frequently lost, while aminoglycoside resistance was generally maintained in the T60 lines (*Figure 3C and D*). Overall, 64.7% of the T60 lines displayed significant decline in resistance to at least one antibiotic, and many displayed loss of resistance to multiple drugs (*Figure 3C and D*, *Figure 3—source data 1*). We found a significant negative correlation between relative fitness and resistance level of T60 lines (Spearman's correlation test, ρ = −0.35, p=0.0031). This indicates that fitness compensation was partly associated with a decline in the original antibiotic resistance level (*Figure 4A and B*, see also *Figure 4—figure supplement 1*, *Figure 4—source data 1*). In summary, approximately 480 generations of evolution in an antibiotic-free medium had a considerable impact on the levels of resistance to multiple antibiotics.

## Phenotypic reversion via compensatory mutations dominates

To gain insights into the underlying molecular mechanisms of resistance loss, 15 independently evolved T60 strains displaying increased fitness were subjected to whole-genome sequencing (see Materials and methods for selection criteria). Using the Illumina platform and established bioinformatics protocols (see Materials and methods), we aimed to identify mutations relative to the genome of the corresponding T0 strains. Altogether, 43 independent mutational events were identified, including 16 single nucleotide polymorphisms (SNPs), 16 deletions and 13 insertions (*Supplementary file 1*). We screened the full bacterial genome to identify resistance-conferring SNPs in the T0 population that revert back to the wild-type sequence in the corresponding T60 strains, but found no such cases. Therefore, the fitness gain in the T60 strains is not due to the molecular reversion of the antibiotic-resistance mutations (*Durão et al., 2018*). Rather, compensatory mutations elsewhere in the genome contribute to the rapid fitness improvement in the evolved strains. A rigorous statistical analysis to test functional relationship between the mutations detected in T0 and T60 was not feasible due to the low number of mutations that have accumulated during the course of laboratory evolution. Nevertheless, we noted several examples on functional relatedness between resistance genes mutated in T0 and genes mutated during lab evolution (i.e. found in T60 strains only, *Figure 5—figure supplement 1*, *Figure 5—source data 1*).

## Pleiotropic side effects of a compensatory mutation in *marR*

Do compensatory mutations simultaneously shape fitness in antibiotic free medium and antibiotic resistance level? To investigate this issue, we have conducted a genetic analysis to explore the potential fitness costs of compensatory mutations. In particular, we studied the impact of selected mutations on growth rate and antibiotic resistance in multiple genetic backgrounds. Several putative compensatory mutations were found to be accumulated in functionally-related transcriptional regulatory proteins involved in anti-drug defense (*Supplementary file 1*). In particular, some of, these proteins control efflux pumps (*marR*, *Alekshun and Levy, 1999*; *Duval et al., 2013*; *Ferenci and Phan, 2015*; *Seoane and Levy, 1995*), lipopolysaccharide biosynthesis (*Seo et al., 2015*), and outer membrane diffusion pores in response to changes in medium osmolarity (*envZ/ompR*, (*Knopp and Andersson, 2015*; *Phan and Ferenci, 2013*).

Here we first focused on a promoter mutation of the *marR* belonging to the mar regulon (*marR\**), not least because this specific gene has clinical relevance: mutations in *marR* have been reported to cause multidrug resistance in clinical *E. coli* isolates in several prior studies (*Komp Lindgren et al., 2003*; *Mazzariol et al., 2000*). This mutation was found in a T60 strain, indicating that it had

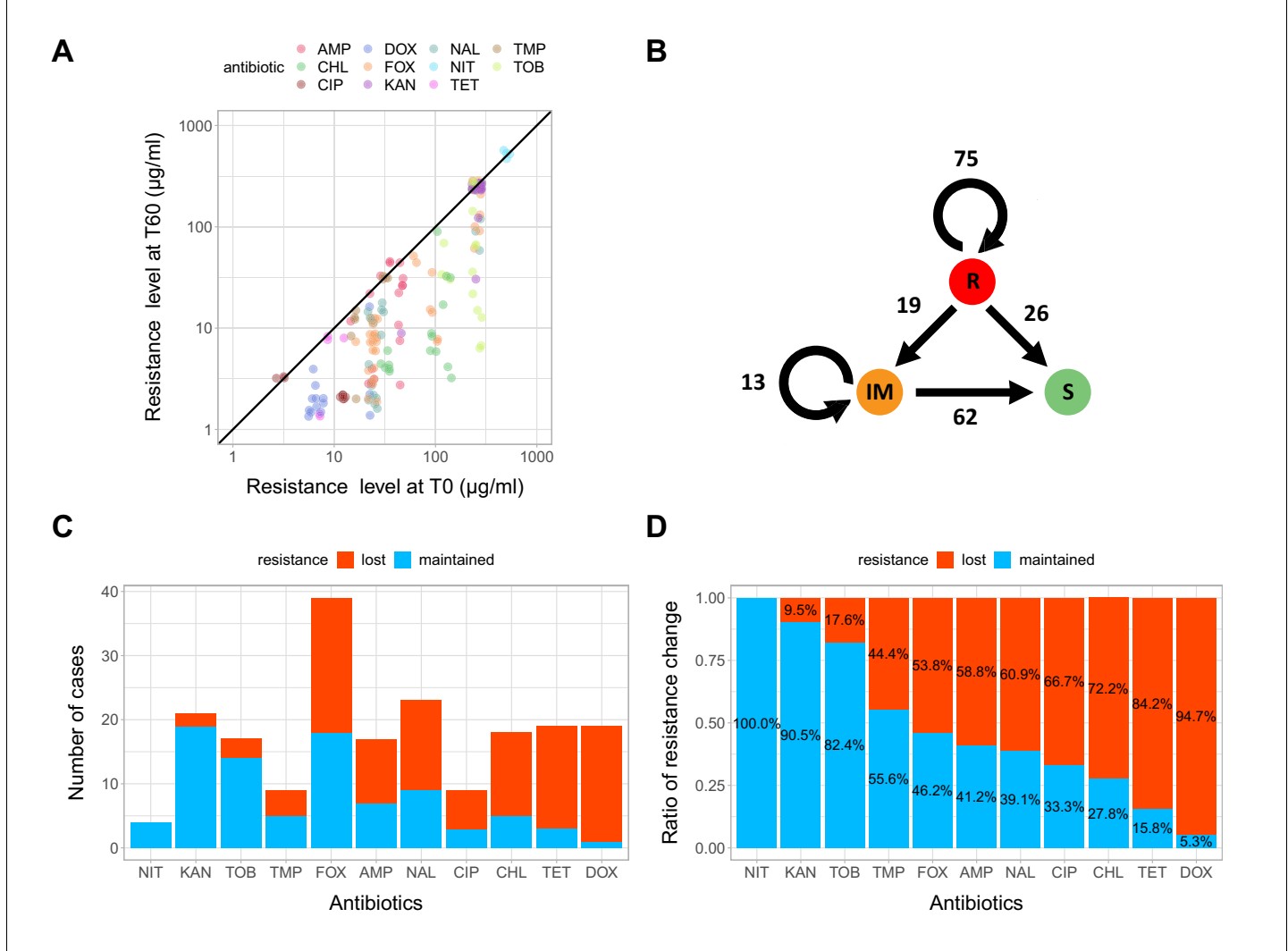

**Figure 3.** Impact of compensatory evolution on antibiotic resistance level. (**A**) The figure depicts minimum inhibitory concentrations (log$_{10}$-scale) of T0 and the corresponding T60 strains. Each of the 195 points represents an antibiotic-strain combination; the colors denote different antibiotics. Points below the black line (x = y) indicates cases with a decrease in antibiotic resistance level after compensatory evolution. For antibiotic abbreviations, see **Table 1**. Source file is available as **Figure 3—source data 1**. (**B**) The figure depicts qualitative changes in resistance level across 195 tested antibiotic-T60 line combinations. Using the CLSI resistance break-point cut-offs, we categorized each T60 and the corresponding T0 strains as being resistant (**R**), intermediate (**IM**) or susceptible (**S**) to each investigated antibiotic. Arrows indicate whether the resistance level was maintained (e.g. R->R) or reduced (e.g. R->S) during the course of laboratory evolution in the antibiotic-free medium. The numbers of antibiotic-T60 line combinations in each category are indicated on the arrows. Source file is available as **Figure 3—source data 1**. (**C and D**) The figures show the number of T60 lines with resistance maintained/lost to the range of antibiotics tested, as absolute number (3C) versus ratio (3D), respectively. Doxycycline (DOX) and tetracycline (TET) resistance was frequently lost (Fisher's Exact test: odds ratio = 0.05, p<0.001 for DOX and odds ratio = 0.20, p<0.01 for TET), while aminoglycoside (kanamycin - KAN, tetracycline - TET) resistance was generally maintained in the T60 lines (Fisher's Exact test: odds ratio = 14.29, p<0.001 for KAN, odds ratio = 6.5, p<0.01 for TOB). For antibiotic abbreviations, see **Table 1**. Source file is available as **Figure 3—source data 1**.
DOI: https://doi.org/10.7554/eLife.47088.008

The following source data is available for figure 3:

**Source data 1.** Changes of cross-resistance interactions following 60 day evolution in an antibiotic free environment.
DOI: https://doi.org/10.7554/eLife.47088.009

accumulated during adaptation to the antibiotic-free environment. This specific mutation (*marR\**) was inserted individually into wild-type *E. coli* and the corresponding antibiotic-resistant T0 strain. First we measured the growth rates of the wild-type and the antibiotic-resistant T0 strain with and without *marR\**. In the absence of antibiotic, the phenotypic effects of this mutation depended on

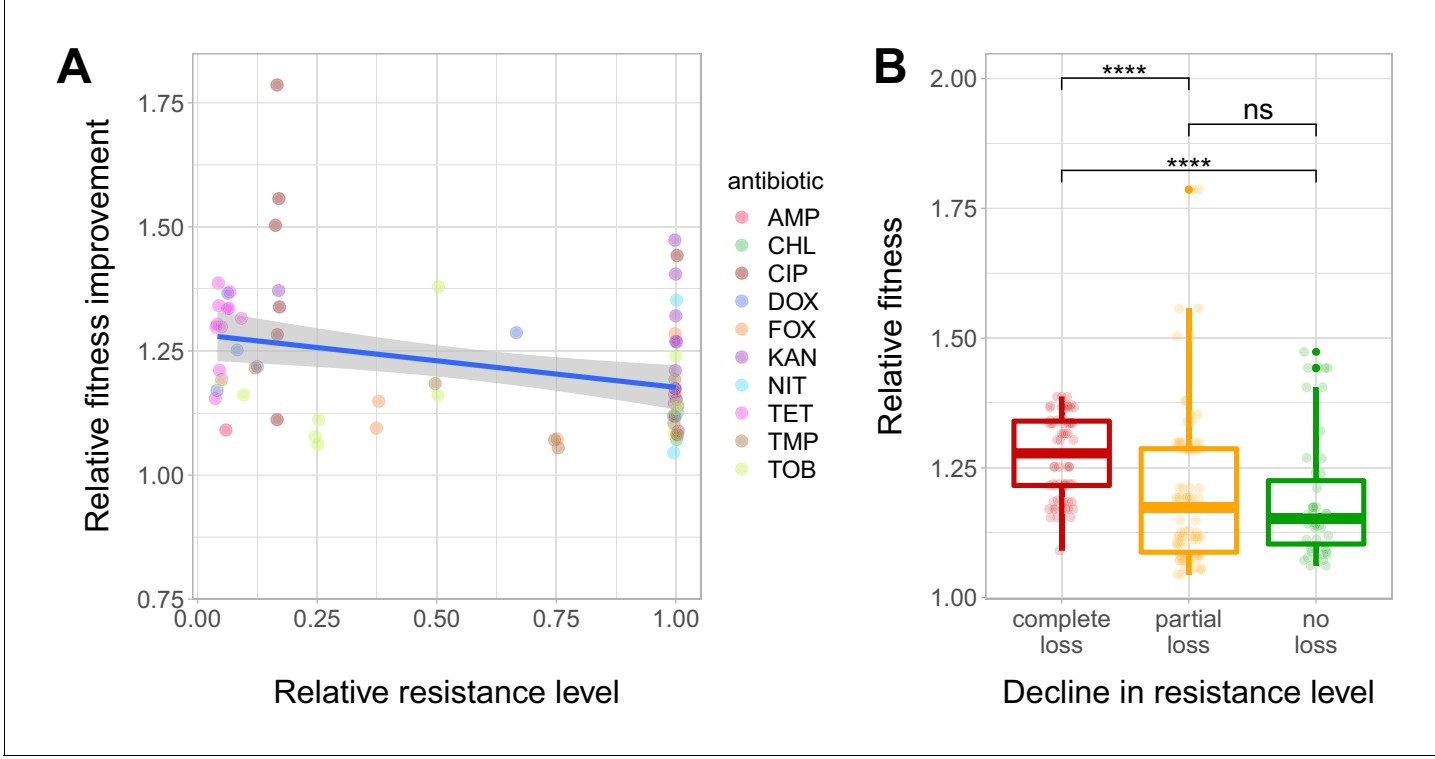

**Figure 4.** Fitness recovery and resistance loss after compensatory evolution. (**A**) The scatterplot shows the relative resistance level and relative fitness improvement of individual T60 strains compared to the corresponding T0 strains (each data point is one strain, the colors indicate the antibiotic used in the analysis). Relative resistance was estimated by the minimum inhibitory concentration of the T60 line relative to that of the T0 line. There is a significant negative correlation between relative fitness improvement and relative resistance level (Spearman's correlation test, ρ = −0.35, p=0.0031). Blue line with gray shaded area represents linear regression line with 95% confidence interval. For antibiotic abbreviations, see **Table 1**. Source file is available as **Figure 4—source data 1**. (**B**) Resistance loss as a function of relative fitness in antibiotic-free environment. The T60 strains were classified into three main categories based on their resistance-profiles: the resistance level declined against all tested antibiotics (complete), declined towards at least one antibiotic (partial), or the resistance level was maintained (no). Using the CLSI resistance break-point cut-offs (CLSI Approved Standard M100, 29th Edition), we categorized each T60 and the corresponding T0 strains as being resistant (**R**), intermediate (**IM**) or susceptible (**S**) to each investigated antibiotic. A decline in resistance was defined by transitions R->IM, R->S or IM->S. We observed a significant association between the relative fitness in the antibiotic-free medium and the decline in resistance level (Mann-Whitney *U*-test: **** indicates p<0.0001, ns indicates that the *p* value is non-significant). Boxplots show the median, first and third quartiles, with whiskers showing the 5th and 95th percentiles. Source file is available as **Figure 4—source data 1**.

DOI: https://doi.org/10.7554/eLife.47088.010

The following source data and figure supplement are available for figure 4:

**Source data 1.** Fitness recovery and resistance loss after compensatory evolution.
DOI: https://doi.org/10.7554/eLife.47088.012
**Figure supplement 1.** Fitness and resistance changes during laboratory evolution.
DOI: https://doi.org/10.7554/eLife.47088.011

the genetic background: it increased growth rate by 4.4% in the corresponding T0 strain, but reduced wild-type fitness (**Figure 5A**, **Figure 5—source data 1**). This epistatic effect suggests that *marR** reduces the deleterious side effects of antibiotic-resistance mutations, while it has a fitness cost in the wild-type strain (**Knopp and Andersson, 2015**; **Praski Alzrigat et al., 2017**). Next, we studied the impact of the *marR** compensatory mutation on resistance level using standard E-test assays. The corresponding T0 strain was found to display detectable resistance to multiple antibiotics, including doxycycline, ampicillin, chloramphenicol, nalidixic acid and tetracycline, while the corresponding T60 strain lost resistance to all studied antibiotics. This can mainly result from the presence of *marR** in the T60 genome, as the introduction of *marR** to the T0 strain recapitulated the same

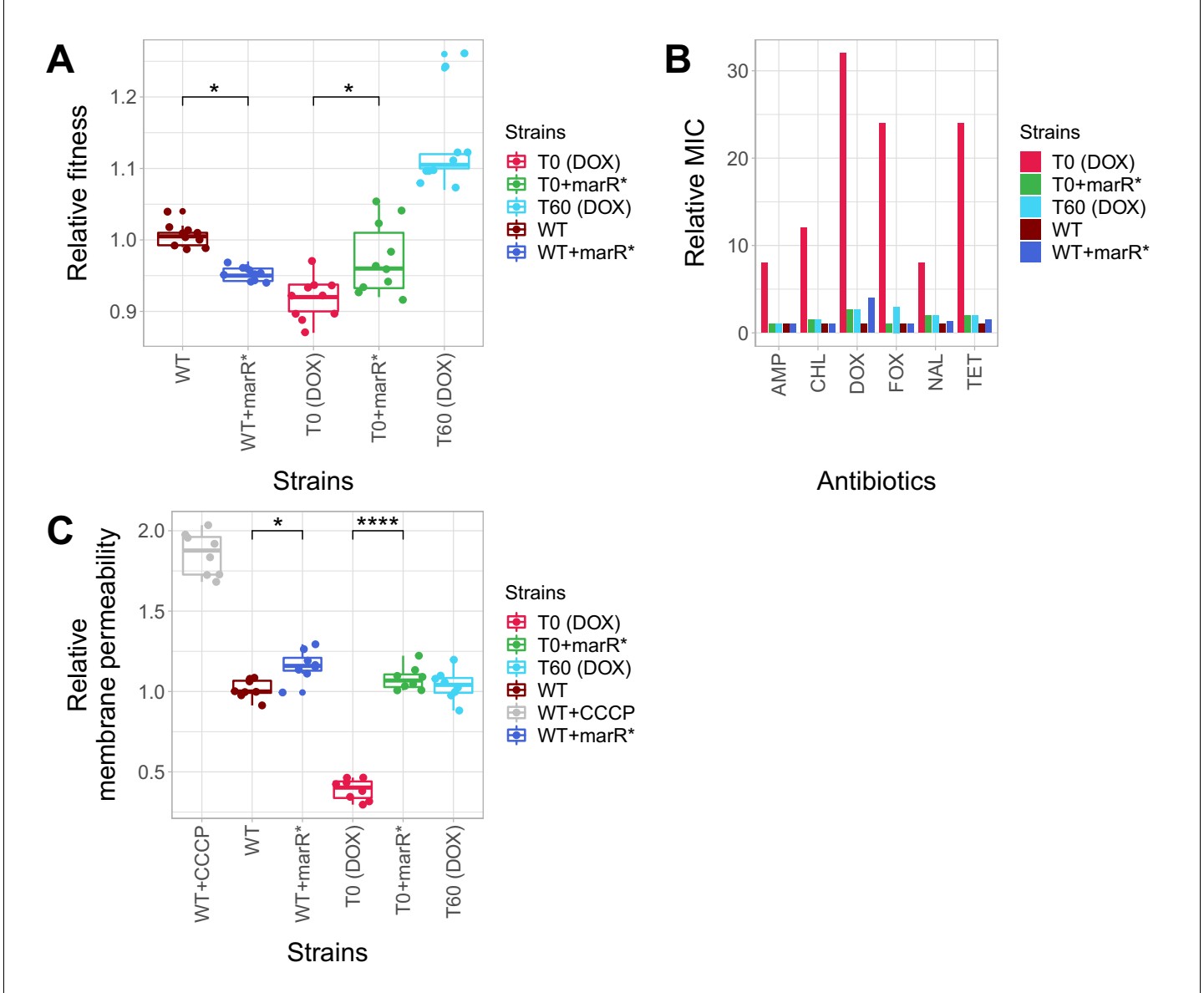

**Figure 5.** Phenotypic effects of a compensatory mutation in the *marR* promoter region. The figure shows the (**A**) relative fitness, (**B**) relative resistance level and (**C**) Relative Hoechst probe accumulation (a proxy of membrane permeability) in the doxycycline-resistant T0 and the corresponding T60 strain harboring a compensatory mutation in the *marR* promoter region (*marR\**). Additionally, *marR\** was introduced into the wild-type and T0 genetic backgrounds as well, yielding WT + marR\* and T0 + marR\* strains, respectively. (**A**) Fitness was measured as the area under the growth curve in an antibiotic-free medium, and was normalized to wild-type fitness. Boxplots show the median, first and third quartiles, with whiskers showing the 5th and 95th percentiles (2 biological and five technical replicates per each genotype). We observed a significant variation in relative fitness across the strains (Tukey's post-hoc multiple comparison tests, * indicates p<0.05). Source file is available as *Figure 5—source data 1*. (**B**) Resistance level of all five strains against six antibiotics. Minimum inhibitory concentration (MIC) was measured by the standard E-test assay, and was normalized to that of the wild-type strain. Only the T0 strain can be considered resistant to each antibiotic tested according to the CLSI resistance break-point cut-off. Source file is available as *Figure 5—source data 1*. (**C**) Membrane permeability across five strains. Membrane permeability was estimated by measuring the intracellular accumulation of a fluorescent probe (Hoechst 33342) in eight biological replicates per each strain or condition. Intracellular accumulation of the probe in the corresponding strains was normalized to that of the wild-type strain. Wild-type cells treated with a protonophore chemical agent (carbonyl cyanide m-chlorophenyl hydrazone, CCCP) served as a positive control, displaying an 88% larger membrane permeability value compared to that of the non-treated wild-type strain. T0 showed an exceptionally low level of Hoechst-dye accumulation compared to all other strains studied, while T0 + marR\* displayed a 166% larger membrane permeability value compared to that of the T0 strain (Tukey's post-hoc multiple comparison tests: **** indicates p<0.0001). Boxplots show the median, first and third quartiles, with whiskers showing the 5th and 95th percentiles. Source file is available as *Figure 5—source data 1*.

DOI: https://doi.org/10.7554/eLife.47088.013

*Figure 5 continued on next page*

*Figure 5 continued*

The following source data and figure supplements are available for figure 5:

**Source data 1.** Relative fitness, relative MIC and relative Hoechst-dye accumulation of reconstructed mutant strains.
DOI: https://doi.org/10.7554/eLife.47088.016
**Figure supplement 1.** Functional relationship of resistance-conferring and compensatory mutations in representative evolved lines.
DOI: https://doi.org/10.7554/eLife.47088.014
**Figure supplement 2.** Effect of a compensatory mutation in *envZ*.
DOI: https://doi.org/10.7554/eLife.47088.015

pattern: the engineered strain lost resistance (*Figure 5B*). Finally, we hypothesized that *marR\** shapes resistance and cellular fitness through antagonistic effects on drug uptake. This hypothesis was tested by measuring the intracellular accumulation of a fluorescent probe (Hoechst 33342) as a proxy for membrane permeability in the resistant T0 strain, in the T60 line and in the wild-type strain with and without the *marR\** compensatory mutation. Decreased intracellular level of the probe indicates either decreased porin activity or enhanced efflux-pump activity (*Coldham et al., 2010*). This is exactly what we found in the resistant T0 strains compared to the wild-type. Importantly, *marR\** restored membrane permeability of T0 to the wild-type level (*Figure 5C*).

In summary, a compensatory mutation in the promoter region of *marR\** increased bacterial fitness in a specific, antibiotic-resistant genotype only. As a side effect, the same mutation increased bacterial susceptibility to multiple antibiotics, probably through elevating membrane permeability. Similar patterns held for a compensatory mutation in *envZ*, a central regulatory protein involved in osmoregulation (*Figure 5—figure supplement 2A and B*, *Figure 5—source data 1*).

## Discussion

It is an open issue whether restricting antimicrobial usage would contribute to the elimination of multidrug-resistant bacteria. Although resistance mutations frequently have associated fitness costs, such costs may decline subsequently through the accumulation of compensatory mutations. It has been argued that such compensatory mutations mitigate the fitness costs of resistance mutations *without* affecting the level of resistance (*Andersson and Hughes, 2010*), suggesting that limiting antibiotic usage may not have much practical utility in clinical settings. However, most prior laboratory studies focused on bacteria carrying a single resistance mutation, whereas antibiotic-resistant clinical isolates usually carry multiple resistance mutations (*Vogwill and MacLean, 2015*). This issue is all the more relevant, as epistasis is prevalent between antibiotic-resistance mutations (*Wong, 2017*).

A specific case of compensation is *molecular reversion*. In this case, the mutation responsible for molecular reversion restores the wild-type, antibiotic-susceptible genetic sequence, and thereby eliminates the fitness costs associated with the resistance mutation. However, molecular reversion is assumed to be generally rare, as it requires very specific and mutational events (*Andersson and Hughes, 2010*; *Durão et al., 2018*). On the other hand, case studies indicate that *phenotypic reversion* can also occur, when the original resistance mutation is maintained, but acquisition of additional mutations simultaneously reduces fitness costs and increases antibiotic-susceptibility. For example, streptomycin-resistance is frequently mediated by resistance mutations in the ribosomal protein gene *rpsL*, but compensatory mutations in other genes involved in translation yield reversion to streptomycin-sensitivity (*Moura de Sousa et al., 2017*). As the molecular targets for phenotypic reversion are relatively broad, reversion to the antibiotic-susceptible state may be far more likely than previously appreciated.

To test the theory of phenotypic reversion, we studied laboratory evolved drug-resistant *E. coli* strains carrying 2 to 13 mutations and initially displaying reduced fitness compared to the wild-type strain. We found that 60 days of laboratory evolution in an antibiotic-free environment led to a rapid fitness improvement in 51% of the antibiotic-resistant lineages (some of which approximated wild-type fitness).

Fitness may increase during the course of laboratory evolution as a result of general adaptation to the environment and/or accumulation of compensatory mutations that mitigate the deleterious side effects of resistance. The second option is more realistic, as several mutations that had accumulated during laboratory evolution affected genes involved in bacterial defense mechanisms against antibiotics or in general stress-responses (eg. *rpoS* promoter region/*nlpD* gene (*Stoebel et al., 2009*), *potD* (*Shah et al., 2011*), *soxSR* (*Jain and Saini, 2016*) (*Supplementary file 1*). An in-depth genetic analysis has also demonstrated epistatic interactions between resistance- and putative compensatory mutations in *marR* and *envZ*. These mutations had accumulated during laboratory evolution and increased fitness in the respective antibiotic-resistant strain, but reduced fitness in the wild-type (*Figure 5A* and *Figure 5—figure supplement 2A*, *Figure 5—source data 1*). This latter finding also suggests that compensatory mutations themselves have associated fitness costs, preventing antibiotic-resistant bacteria to reach the full fitness of sensitive variants.

Crucially, we have demonstrated that drug-resistance declines in an antibiotic-free laboratory environment. In as few as 480 generations, 64.7% of drug-resistant *E. coli* strains showed elevated susceptibilities to at least one antibiotic investigated (*Figure 3—source data 1*). We did not observe *bona fide* reversion mutations in laboratory evolved bacteria (*Durão et al., 2018*). This is not unexpected, as compensation via the accumulation of additional mutations elsewhere in the genome is far more likely to occur.

Detailed genetic analysis of the *mar* regulon also supports the phenotypic reversion hypothesis. MarR is a transcriptional regulatory protein that controls the activity of the *mar* regulon in *E. coli* through the repression of *marA*. The *mar* regulon participates in controlling several genes involved in antibiotic-resistance, including the AcrA/AcrB/TolC multidrug-efflux system (*Figure 6*). In response to antibiotic stresses (e.g. doxycycline or ciprofloxacin), *marR* is regularly mutated both in clinical and in laboratory settings, leading to increased expression of *marA* and other members of the *mar* regulon (*Praski Alzrigat et al., 2017*). Here we have focused on a multidrug-resistant laboratory evolved *E. coli* strain that carries a mutation in the protein coding sequence of *marR*. This resistance mutation has an associated fitness cost (*Figure 5—source data 1*), promoting the accumulation of further mutations. Our study indicates that this can be achieved by a compensatory mutation in the promoter region of the *mar* operon. This compensatory mutation increases bacterial fitness, susceptibility to multiple antibiotics alike, and restores wild-type-like membrane permeability, probably through changing the activity of the *mar* regulon. In a follow-up work we are planning to study this phenomenon in detail.

It is important to emphasize that loss of antibiotic resistance is not equally likely across antibiotic-resistant strains. For example, in our study, resistance to doxycycline and tetracycline was frequently lost, while aminoglycoside-resistance was generally maintained during the course of laboratory evolution (*Figure 3C and D*).

Our findings appear to be consistent with clinical data. For instance, a Finnish retrospective study assessed the proportion of quinolone-susceptible *E. coli* urine isolates before and after a nationwide restriction of ciprofloxacin use was implemented in Finland. The research revealed that a reduced consumption of quinolone antibiotics resulted in a significant decrease in quinolone-resistance of *E. coli* (*Gottesman et al., 2009*). Our laboratory study also shows that ciprofloxacin-resistance has declined in 66% of initially resistant populations, following a long-term exposure to antibiotic-free medium (*Figure 3D*). Another study examined the impact of a 24 month voluntary restriction on the use of trimethoprim-containing drugs in Sweden on the prevalence of trimethoprim-resistant *E. coli* isolated from urinary-tract infections. All clinical isolates were found to retain their resistance levels and carried mutation in *folA*, the target gene of trimethoprim, even after 24 months of trimethoprim restriction (*Brolund et al., 2010*). In agreement with this clinical study, we have found that all five trimethoprim resistant *E. coli* strains with a *folA* resistance mutation have maintained their resistance following laboratory evolution in antibiotic-free medium (*Figure 3*-source data, *Figure 4—source data 1*). These considerations must be taken with some caution, as comparison of clinical and laboratory data is not straightforward. For instance, restricted usage of certain antibiotics in hospitals cannot completely eliminate antibiotic selection in a given region due to lack of isolation and cross-resistance between antibiotics.

In summary, three main patterns indicate that phenotypic reversion to an antibiotic-susceptible state could be common during compensatory evolution. In our study, we have observed i) rapid

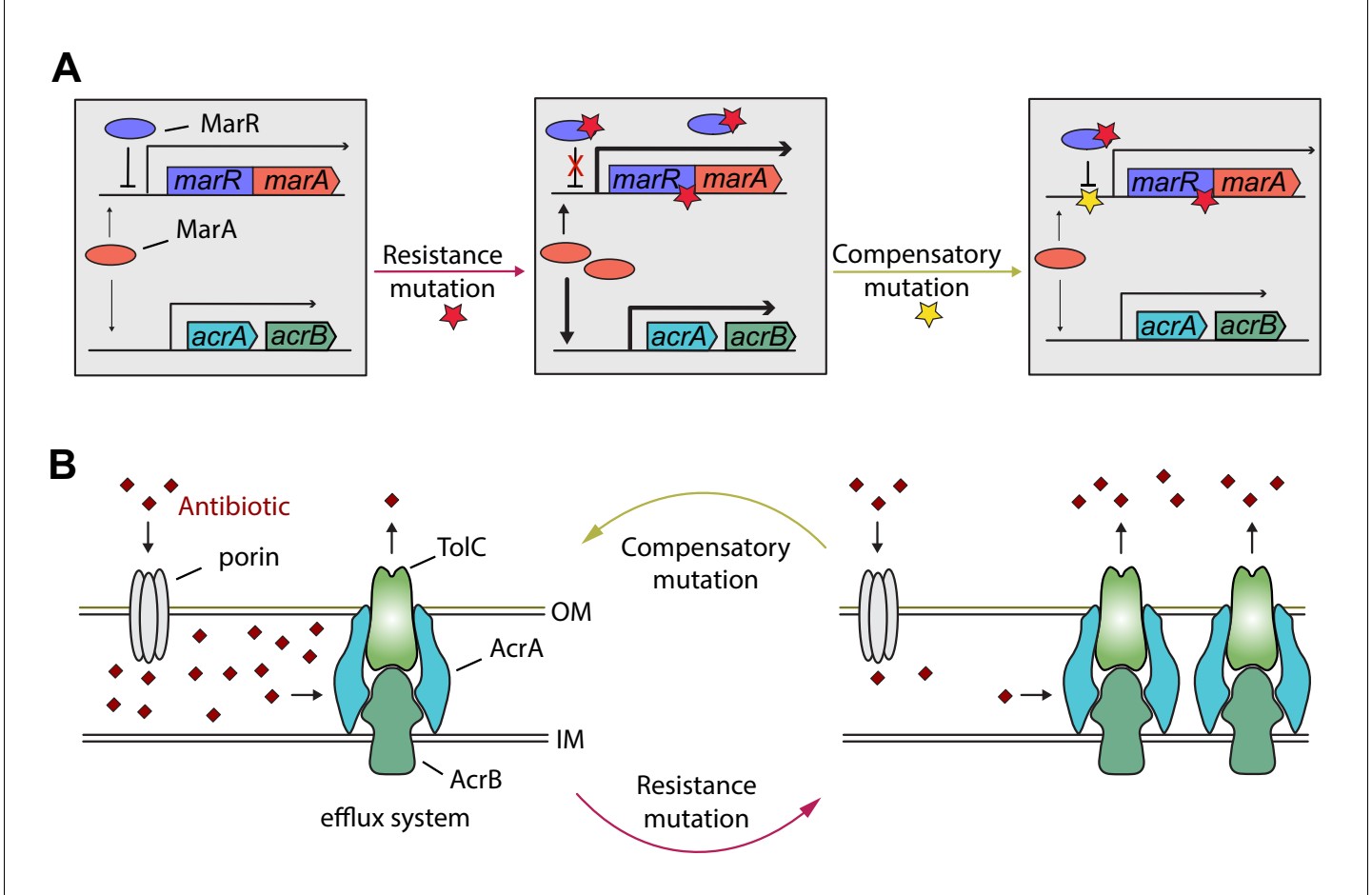

**Figure 6.** Hypothetical mechanism of compensation by a *marR* compensatory mutation. The *mar* regulon participates in controlling several genes involved in resistance to antibiotics including the AcrA/AcrB/TolC multidrug efflux system (panel A). MarR is a transcriptional regulatory protein that controls the activity of the *mar* system in wild-type *E. coli* through the repression of *marA*. In response to antibiotic stresses (e.g. doxycycline or ciprofloxacin), *marR* is mutated (indicated by a red star), leading to increased expression of *marA* and, subsequently, other members of the *mar* regulon (*Praski Alzrigat et al., 2017*). However, the elevated activity of the *mar* regulon is harmful in antibiotic-free conditions, promoting the accumulation of further mutations. Our study indicates that this can be achieved by a compensatory mutation in the promoter region of the *mar* operon (indicated by a yellow star). The compensatory mutation putatively restores the activity of the *mar* regulon to the wild-type level (panel B). Source file is available as *Supplementary file 1*.

DOI: https://doi.org/10.7554/eLife.47088.017

fitness increase in antibiotic free-medium, ii) associated loss of antibiotic resistance, and we iii) identified specific mutations that simultaneously change both characteristics.

Our findings suggest that restricting antimicrobial usage could be a useful policy to control the increasing rise and spread of multidrug-resistant bacteria, but it may work for certain antibiotics only. We should also emphasize, however, that all our evolutionary experiments were performed in devoid of *any* antibiotics. The next logical step is to study the occurrence of phenotypic reversion during exposure to temporarily changing antibiotic treatments or sublethal antibiotic dosages as it may occur in real-life situations (*Andersson and Hughes, 2014*). Our study leaves open the question about the extent by which initial fitness costs of resistance mutations and/or intensity of antibiotic selection shapes subsequent compensatory evolution and associated loss of resistance. Finally, our study has focused on chromosomal resistance mutations. It is still unclear whether resistance-conferring plasmids are generally lost or the associated fitness costs are mitigated by genomic mutations (*Dahlberg and Chao, 2003*).

# Materials and methods

## Key resources table

| Reagent type (species) or resource | Designation | Source or reference | Identifiers | Additional information |
|---|---|---|---|---|
| Strain, strain background (*Escherichia coli*) | K12 BW25113 (wild type) | PMID: 25000950 | | SRA project: SRX551437 (*E. coli* AB cross-resistance) |
| Strain, strain background (*Escherichia coli*) | SRR1297006 to SRR1297184 (resistant strains) | PMID: 25000950 | | SRA project: SRX551437 (*E. coli* AB cross-resistance) |
| Chemical compound, drug | ETEST AMPICILLIN AM 256 | bioMérieux | S30-412253 | |
| Chemical compound, drug | ETEST CEFOXITIN FX 256 | bioMérieux | S30-412285 | |
| Chemical compound, drug | ETEST CIPROFLOXACIN CI 32 | bioMérieux | S30-412311 | |
| Chemical compound, drug | ETEST CHLORAMPHENICOL CL 256 | bioMérieux | S30-412309 | |
| Chemical compound, drug | ETEST DOXYCYCLINE DC 256 | bioMérieux | S30-412328 | |
| Chemical compound, drug | ETEST KANAMYCIN KM 256 | bioMérieux | S30-412382 | |
| Chemical compound, drug | ETEST NALIDIXIC-ACID NA 256 | bioMérieux | B30-516500 | |
| Chemical compound, drug | ETEST NITROFURANTOIN NI 512 | bioMérieux | B30-530400 | |
| Chemical compound, drug | ETEST TETRACYCLINE TC 256 | bioMérieux | S30-412471 | |
| Chemical compound, drug | ETEST TOBRAMYCIN TM 256 | bioMérieux | S30-412479 | |
| Chemical compound, drug | ETEST TRIMETHOPRIM TR 32 | bioMérieux | S30-412483 | |
| Chemical compound, drug | bisBenzimide H 33342 trihydrochloride | Sigma-Aldrich | B2261-25MG | PMID: 20513705 |
| Chemical compound, drug | carbonyl cyanide-m-chlorophenyl hydrazone (CCCP) | Life Technologies | AC228131000 | PMID: 20513705 |
| Recombinant DNA reagent | pORTMAGE-3 | PMID: 26884157 | Addgene: Plasmid #72678 | |
| Software, algorithm | RStudio (Version 1.1.463) | R Core Team (2018). R: A language and environment for statistical computing. R Foundation for Statistical Computing, Vienna, Austria | | http://www.r-project.org/ |
| Software, algorithm | ggplot2 package (version 3.0.0) | H. Wickham. ggplot2: Elegant Graphics for Data Analysis. Springer-Verlag New York, 2016. | | https://cran.r-project.org/web/packages/ggplot2/ggplot2.pdf |

## Strains

### Wild-type strain

*Escherichia coli* K12 BW25113 was used as a wild-type control in all experiments.

### Antibiotic-resistant strains

The 60 multidrug-resistant strains used in this study were derived from our previous work (*Lázár et al., 2014*), where parallel evolving populations of *E. coli* K12 BW25113 were adapted to

increasing dosages of one of 12 antibiotics (*Figure 1—figure supplement 1*). These antibiotics employed were the following: ampicillin (AMP), cefoxitin (FOX), chloramphenicol (CHL), ciprofloxacin (CIP), doxycycline (DOX), erythromycin (ERY), kanamycin (KAN), nalidixic acid (NAL), nitrofurantoin (NIT), tetracycline (TET), trimethoprim (TMP) and tobramycin (TOB). The evolutionary experiment was continued for ~240–384 generations, at which point the evolving populations reached an up to 328-fold increase in resistance compared to the wild-type ancestor. In all cases (except for the CPR9 strain which was found to be characterized by an intermediate level of resistance), the resistance levels were above the current clinical breakpoints for resistance according to the Clinical and Laboratory Standards Institute (CLSI Approved Standard M100, 29th Edition) guidelines. In spite of single antibiotic pressure, the evolution of multidrug-resistance was a frequent phenomenon. The 60 antibiotic-resistant strains (4–6 strains per antibiotic) were previously subjected to whole-genome sequencing. The identified resistance mutations affected drug targets, cell permeability or efflux pumps.

## Medium

Unless otherwise stated, we used minimal salts (MS) medium (1 g/L $(NH_4)_2SO_4$, 3 g/L $KH_2PO_4$ and 7 g/L $K_2HPO_4$) supplemented with 1.2 mM $NA_3C_6H_5O_7 \times 2H_2O$, 0.4 mM $MgSO_4$, 0.54 µg/mL $FeCl_3$, 1 µg/mL thiamine hydrochloride, 0.2% Casamino-acids and 0.2% glucose. As an exception, Lysogeny Broth (LB containing 10 g tryptone, 5 g yeast extract, 5 g sodium chloride per 1 L of water) and Terrific Broth (TB containing 24 g yeast extract, 12 g tryptone, 9.4 g $K_2HPO_4$, 2 g $KH_2PO_4$ per 1 L of water) were applied during the pORTMAGE protocol. All components were obtained from Sigma-Aldrich.

## Antibiotics

In order to measure the resistance profile of the T60 and corresponding T0 evolved lines we used 11 of the above-mentioned 12 antibiotics (*Table 1*). Erythromycin (ERY) was excluded from the analysis, as none of the studied strains displayed cross-resistance to it. Standard E-test strips for all remaining antibiotics were purchased from bioMérieux. Powder stocks of antibiotics were purchased from Sigma-Aldrich, except for DOX (AppliChem). Antibiotic solutions were freshly prepared on a weekly basis from powder stocks, kept at –20°C and were filter-sterilized before use.

## Laboratory evolutionary experiment

The laboratory evolution experiment followed an established protocol (*Lázár et al., 2014*). Briefly, we started with 23 antibiotic-resistant *E. coli* strains that displayed a significant fitness cost. We excluded erythromycin- (ERY) and nalidixic acid- (NAL) resistant strains from the evolutionary experiment, as these strains did not show a significant fitness cost. six parallel lines were initiated from each antibiotic-resistant strain, and were propagated in 96-well microtiter plates in antibiotic-free MS medium for 60 days. All parallel lines were inoculated into randomly selected positions of these 96-well plates. The plates also contained control wells in several positions that were not inoculated by cells to help plate identification and orientation as well as to avoid cross-contamination of parallel evolving lines. Using a manual-held 96-pin replicator (VP407, V and P Scientific), roughly 1.2 µl of each stationary phase culture was transferred every day to 100 µl of fresh medium. At every 120 transfers, a fraction of the overnight culture was kept at –80°C as a glycerol stock. Cross-contamination events were regularly checked by visual inspection of empty wells.

## High-throughput fitness measurements and determination of growth rate

### Fitness measurements

Established protocols were used to measure fitness in bacterial lines (*Warringer and Blomberg, 2003*). Starter cultures were inoculated from frozen samples into 96-well plates. The starter plates were grown for 24 hr under conditions identical to the evolutionary experiment. 384-well plates filled with 60 µl MS minimal medium per well were inoculated for growth curve recording compared to the starter plates, using a pintool with 1.58 mm floating pins. The pintool was moved by a Microlab Starlet liquid handling workstation (Hamilton Bonaduz AG) to provide uniform inoculum across all samples. The 384-well plates were incubated at 30°C in an STX44 (LiCONiC AG) automated

incubator with alternating shaking speed every minute between 1,000 rpm and 1,200 rpm. Plates were transferred by a Microlab Swap 420 robotic arm (Hamilton Bonaduz AG) to Powerwave XS2/HT plate readers (BioTek Instruments Inc) every 20 min and cell growth was followed by recording the optical density at 600 nm. Five technical replicates of three biological replicate measurements were executed on all strains sampled from each time-point of the evolutionary experiment.

## Growth curve analysis

Fitness was approximated by calculating the area under the growth curve (AUGC). AUGC has been previously used as a proxy for fitness (*Hasenbrink et al., 2005*) and it has the advantage to integrate multiple fitness parameters, such as the slope of exponential phase (growth rate) and the final optical density (yield). AUGC was calculated from the obtained growth curves of a 1000 min time interval following the end of lag phase. The end of the lag phase was identified according to an established protocol (*Warringer and Blomberg, 2003*; *Warringer et al., 2003*). To eliminate potential within-plate effects that might cause measurement bias, relative fitness was normalized by the fitness of the neighboring reference wells that contained wild-type controls. For each line and each evolutionary time point, relative fitness was calculated as the median of the normalized AUGC of the technical replicates divided by the median fitness of the wild-type controls. At day 0, the technical replicate measurements of the isogenic, independently evolving lines were combined to calculate median fitness for the ancestral antibiotic-resistant strain, since at that time these populations had no independent evolutionary history. Stringent criteria were used to define the set of antibiotic-resistant strains with a substantial fitness defect: significance was determined by the Mann-Whitney *U*-test. Significance of fitness increase for the evolving lines derived from antibiotic-resistant strains having initial fitness defect was also calculated by the Mann-Whitney U-test. All tests were corrected for false discovery rate (FDR-corrected *p* value < 0.05).

As expected, fitness estimated by AUGC showed a significant positive correlation with fitness estimated by growth rate (Pearson's correlation, r = 0.41, p<2.2e-16) or yield (Pearson's correlation, r = 0.78, p<2.2e-16). Additionally, we found that AUGC is more robust than yield, that is it shows less variation across biological replicates (median CVs values for AUGC and yield are 8.3% and 14.1%, respectively).

## Determination of the minimal inhibitory concentration (MIC)

Minimal inhibitory concentrations (MICs) were determined using standard E-test strips (bioMérieux) according to the manufacturer's instructions. Briefly, overnight cultures of bacteria were diluted to an optical density ($OD_{600}$) of 0.6. 100 µl of the diluted inoculum was spread on each MS agar plate and the plates were incubated at 30°C for 24–48 hr. MICs were read directly from the E-test strips according to the instructions of the manufacturer. Based on the MIC results, we categorized each strain as being resistant (R), intermediate (IM) or susceptible (S) to each investigated antibiotic according to the Clinical and Laboratory Standards Institute (CLSI Approved Standard M100, 29th Edition) guidelines.

## Hoechst 33342 (Bisbenzimide H 33342) accumulation assay

To estimate changes in membrane permeability and efflux pump activity, we implemented a scalable fluorescence assay (*Coldham et al., 2010*). This method is based on the intracellular accumulation of the fluorescent probe Hoechst 33342 (Bisbenzimide H 33342, Sigma-Aldrich). Strains were cultured in eight biological replicates overnight at 30°C and stationary phase cultures were regrown in fresh medium to an optical density ($OD_{600}$) of 0.6 at 30°C. Bacterial cells were collected by centrifugation at 4000 g and resuspended in 1 mL phosphate-buffered saline (PBS). The optical density ($OD_{600}$) of all suspensions was adjusted to 0.1, and 0.18 mL of each suspension was transferred to 96-well plates (CellCarrier-96 Black Optically Clear Bottom, supplied by Sigma-Aldrich). Plates were incubated in a Synergy two microplate reader at 30°C, and 25 µM Hoechst 33342 was added to each well. The wild-type *E. coli* K12 BW25113 treated with an efflux inhibitor agent (carbonyl cyanide-m-chlorophenyl hydrazone, CCCP, Life Technologies) served as a positive control. The $OD_{600}$ and fluorescence curves were recorded for 1 hr with 75 s delays between readings. Fluorescence was read from the top of the wells using excitation and emission filters of 355 and 460 nm, respectively. The first 15 min were excluded from further analysis due to the high standard deviation between

replicates. Blank normalized $OD_{600}$ values were calibrated by applying the following transformation: $OD_{calibrated} = OD_{600} + 0.49312 * OD_{600}^3$. Data curves were smoothed and fluorescence per $OD_{600}$ ratio curves was calculated. Finally, areas under these ratio curves were determined.

## Allelic replacements

Utilizing multiplex automated genome engineering (MAGE) (*Nyerges et al., 2016*) we reconstructed two candidate T60 compensatory mutations and the corresponding T0 resistance mutations in the wild-type genetic background (separately and in combination as well), as well as the compensatory mutations in the corresponding initial antibiotic-resistant strains.

Each MAGE cycle consisted of the following steps: Upon reaching $OD_{600} = 0.4$–$0.6$, cells were transferred to a 42°C shaking water bath to induce λ-Red protein expression for 15 min at 250 rpm. Cells were then immediately chilled on ice for at least 10 min. Electrocompetent cells were made by washing and pelleting the cells twice in 10 mL of ice-cold $dH_2O$. 40 μL cell suspension was mixed with 1 μL of 100 μM oligonucleotide. Electroporation was done on a BTX (Harvard Apparatus) CM-630 Exponential Decay Wave Electroporation System in 1 mm gap VWR Signature Electroporation cuvettes (1.8 kV, 200 Ω, 25 μF). Immediately after electroporation, 1 mL TB + 2 ml LB medium was added onto the cells to allow recovery. The 100,000 diluted cells were spread onto solid medium and incubated at 30°C for 24 hr. Allelic replacement frequencies for all strains were measured at each locus by allele-specific PCR. Selected clones carrying the desired modifications were verified by capillary sequencing.

## Whole-genome sequencing

To identify potential compensatory mechanisms, 15 T60 adapted lines derived from a total of 10 antibiotic-resistant T0 strains were chosen for whole-genome sequencing. These T60 lines were chosen to represent diverse patterns of resistance loss, as well as to cover T0 strains adapted to a variety of antibiotics. *E. coli* genomic DNA was prepared with GenElute Bacterial Genomic DNA Kit (Sigma-Aldrich) and quantified using Qubit dsDNA BR assay in a Qubit 2.0 fluorometer (Invitrogen). 200 ng of genomic DNA was fragmented in a Covaris M220 focused-ultrasonicator (peak power: 55W, duty factor: 20%, 200 cycles/burst, uration: 45 s) using Covaris AFA screw cap fiber micro-TUBEs. Fragment size distribution was analyzed by capillary gel electrophoresis using Agilent High Sensitivity DNA kit in a Bioanalyzer 2100 instrument (Agilent) and then indexed sequencing libraries were prepared using TruSeq Nano DNA LT kit (Illumina) following the manufacturer's instructions. This, in short, includes end repair of DNA fragments, fragment size selection, ligation of indexed adapters and library enrichment with limited-cycle PCR. Sequencing libraries were validated (library-sizes determined) using Agilent High Sensitivity DNA kit in a Bioanalyzer 2100 instrument, then quantified using qPCR based NEBNext Library Quant Kit for Illumina (New England Biolabs) with a Piko-Real Real-Time PCR System (Thermo Fisher Scientific) and diluted to 4 nM concentration. Groups of 12 indexed libraries were pooled, denatured with 0.1 N NaOH, and after dilution loaded in a MiSeq Reagent kit V2-500 (Illumina) at 8 pM concentration. 2 × 250 bp pair-end sequencing was done with an Illumina MiSeq sequencer, primary sequence analysis was done on BaseSpace cloud computing environment with GenerateFASTQ 2.20.2 workflow.

Paired-end sequencing data were exported in FASTQ file format. The reads were trimmed using Trim Galore (Babraham Bioinformatics) and cutadapt (*Martin, 2011*) to remove adapters and bases where the PHRED quality value was less than 20. Trimmed sequences were removed if they became shorter than 150 bases. FASTQC program (https://www.bioinformatics.babraham.ac.uk/projects/fastqc/) was used to evaluate the quality of the original and trimmed reads.

The Breseq program was used with default parameters for all samples. The gdtools was used for annotating the effects of mutations and compare multiple samples. The genbank formatted reference genome BW25113.gb was used as a reference genome in the analysis. Sequencing data have been deposited in the NCBI Sequence Read Archive (SRA) under the accession number of PRJNA529335: (URL: https://www.ncbi.nlm.nih.gov/sra/PRJNA529335).

# Additional information

## Funding

| Funder | Grant reference number | Author |
|---|---|---|
| H2020 European Research Council | H2020-ERC-2014-CoG 648364 – Resistance Evolution | Csaba Pál |
| Gazdaságfejlesztési és Innovációs Operatív Programm | GINOP-2.3.2-15-2016-00014 | Csaba Pál |
| Gazdaságfejlesztési és Innovációs Operatív Programm | GINOP-2.3.2-15-2016-00020 | Csaba Pál |
| Hungarian Academy of Sciences | Momentum Programme LP-2017-10/2017 | Csaba Pál |
| National Research, Development and Innovation Office | Élvonal Programme KKP 126506 | Csaba Pál |
| Wellcome Trust | WT 098016/Z/11/Z | Balázs Papp |
| Gazdaságfejlesztési és Innovációs Operatív Programm | GINOP-2.3.2-15-2016-00026 | Balázs Papp |
| Wellcome Trust | WT 084314/Z/07/Z | Csaba Pál |
| National Research, Development and Innovation Office | FK 128775 | Zoltán Farkas |
| Magyar Tudományos Akadémia | Lendület Programme LP 2012-32/2018 | Csaba Pál |
| Magyar Tudományos Akadémia | Postdoctoral Programme PD-007/2016 | Viktória Lázár |
| Magyar Tudományos Akadémia | Postdoctoral Programme PD-038/2015 | Zoltán Farkas |
| National Research, Development and Innovation Office | NKFI-112294 | László Bodai |
| Magyar Tudományos Akadémia | Lendület Programme LP2009-013/2012 | Balázs Papp |

The funders had no role in study design, data collection and interpretation, or the decision to submit the work for publication.

## Author contributions

Anett Dunai, Zoltán Farkas, Conceptualization, Formal analysis, Investigation, Visualization, Methodology, Writing—original draft, Writing—review and editing, Interpretation of data; Réka Spohn, Conceptualization, Formal analysis, Supervision, Investigation, Methodology, Writing—original draft; Viktória Lázár, Conceptualization, Formal analysis, Supervision, Investigation, Methodology; Ádám Györkei, Balázs Szappanos, Formal analysis, Interpretation of data; Gábor Apjok, Methodology; Gábor Boross, Formal analysis, Methodology, Interpretation of data; Gábor Grézal, Formal analysis, Visualization, Interpretation of data; Anikó Faragó, László Bodai, Resources, Bioinformatic analysis of whole genome sequencing data; Balázs Papp, Resources, Funding acquisition, Writing—review and editing; Csaba Pál, Conceptualization, Supervision, Funding acquisition, Writing—original draft, Writing—review and editing

## Author ORCIDs

Anett Dunai (iD) https://orcid.org/0000-0001-6270-9603
Zoltán Farkas (iD) https://orcid.org/0000-0002-5085-3306
Gábor Boross (iD) https://orcid.org/0000-0002-7208-5678
László Bodai (iD) https://orcid.org/0000-0001-8411-626X
Csaba Pál (iD) https://orcid.org/0000-0002-5187-9903

**Decision letter and Author response**
Decision letter https://doi.org/10.7554/eLife.47088.023
Author response https://doi.org/10.7554/eLife.47088.024

## Additional files

### Supplementary files
• Supplementary file 1. Mutations detected by whole genome sequencing. This file contains the mutations acquired during 60 day evolution in an antibiotic free environment. This file was used to create *Figure 5—figure supplement 1* and *Figure 6*.
DOI: https://doi.org/10.7554/eLife.47088.018
• Transparent reporting form
DOI: https://doi.org/10.7554/eLife.47088.019

### Data availability
Sequencing data have been deposited in the NCBI Sequence Read Archive (SRA) under the accession number of PRJNA529335.

The following dataset was generated:

| Author(s) | Year | Dataset title | Dataset URL | Database and Identifier |
|---|---|---|---|---|
| Dunai A, Spohn R, Lázár V, Györkei V, Farkas Z, Apjok G, Boross G, Szappanos B, Grézal G, Faragó A, Bodai L, Papp B, Pál C | 2019 | WGSS of antibiotic resistant *E. coli* strains evolved on antibiotic-free medium | https://www.ncbi.nlm.nih.gov/sra/PRJNA529335 | NCBI Sequence Read Archive, PRJNA529335 |

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
