## [Decision Letter]

Thank you for submitting your article "Rapid decline of bacterial multidrug-resistance in an antibiotic-free environment through phenotypic reversion" for consideration by *eLife*. Your article has been reviewed by three peer reviewers, one of whom is a member of our Board of Reviewing Editors, and the evaluation has been overseen by a Reviewing Editor and Patricia Wittkopp as the Senior Editor. The reviewers have opted to remain anonymous.

The reviewers have discussed the reviews with one another and the Reviewing Editor has drafted this decision to help you prepare a revised submission.

Summary:

The authors examine the loss of antibiotic resistance after the termination of experimental evolution that lead to adaptation and increased antibiotic resistance. They examine the cost of resistance and the role of compensatory mutations (or reversions) in alleviating this cost. The three reviewers appreciated the quality of the work and its general interest, but they raised a number of major concerns that can be addressed by modifying the text, reanalyzing some of the results and potentially adding relatively minor experiments. Some of these major concerns relate to the descriptive and qualitative nature of some of the analyses shown that are critical to supporting the conclusions of the study. Others relate to the quality of the presentation and of the discussion of the results as they relate to previous work or to the limitation of the study.

Essential revisions:

1) The manuscript is generally well written and referenced satisfyingly. However, the introduction is somewhat lagging behind the rest of the text. Claims in the introduction are not always supported by appropriate citations and quality of the writing could be improved in this section. The authors are referred to some specific comments below for examples of elements of the introduction that could be improved. For instance, the introduction describes an important issue related to the fitness cost of antimicrobial resistance, which is the disconnect between the results of experimental evolution studies and clinical observations. Whether resistance to antimicrobials really does carry a fitness cost, and whether it may or may not be maintained in populations of pathogens thanks to compensatory evolution is indeed controversial, and contradictory evidence currently prevents a definitive answer to this question. This manuscript does not directly address this issue, yet the introduction would suggest otherwise. The authors are invited to revise the introduction to better expose the focus and scope of the study.

2) Results on the loss of resistance appear overly qualitative, at least in the format used in Figure 3 and in the text (subsection “Rapid loss of multi-drug resistance in an antibiotic-free medium”). We believe a synthetic quantitative analysis may be applied to the results provided in Supplementary file 3 and presented in a visually compelling manner. A quantitative measure of loss in resistance and its causal link or not with loss of cost would be more convincing, and would help readers grasp the magnitude of this phenotypic change. The authors already have MIC data: it is just a matter of using it in a transparent data visualization format, which may still use the qualitative categories, but perhaps as an added layer on top of the quantitative data, rather than as the sole reported measure.

3) As presented, the results suggest association between fitness recovery and resistance loss. The manuscript would be enhanced by a more formal statistical analysis of this association.

4) The Discussion section provides a list of evidence that the authors claim supports a driving role for compensatory evolution in adaptation to the antibiotic-free environment. We think that 2 out of the 3 evidences mentioned does not support that claim. The first reads: "First, our analysis has revealed major differences in relative fitness across strains adapted to different antibiotics." We fail to see how differences in fitness between strains supports any role for compensatory evolution. It rather suggests a diversity of molecular mechanisms, which may or may not involve compensation. In other words, this phenomenon appears unrelated to compensation. The next claim which we believe is not supported by the stated evidence is the following: "Third, an in-depth genetic analysis has demonstrated that mutations in marR and envZ increased fitness in the respective antibiotic-resistant strain, but reduced fitness in the wild-type". This is evidence for epistasis, but not for compensation per se.

5) In comparison to the extensive analysis done in Lazar Nat Communications 2014, I would have preferred a similar detailed analysis of the evolutionary experiments in this follow-up work. The broad conclusions are expected, but there are insights which this work could uniquely address. For one, it is interesting to know how the fitness cost and resistance level vary with generation-while there is a broad phenotypic reversion of fitness cost, does this happen early on the evolution, does this reversion occur at the same time that resistance is reacquired (or not)? Additionally, what is the reproducibility/repeatability of these time courses across the replicates?

6) I am curious to know when the mutations found at the end of the experiment (listed in Supplementary file 4) occurred during the lab evolution. Moreover, do the occurrence of specific mutations correlate with the time-changes of fitness cost and resistance level throughout the evolution? This is particularly crucial because there are multiple mutations at the end of the lab evolution and correlating these with fitness cost and resistance level could determine causality and epistasis between mutations.

7) Why do many of the T60 lines have worse fitness after evolution in antibiotic-free medium? One would expect the fitness to at least stay the same or improve.

8) Subsection “Laboratory evolution of antibiotic-resistant strains in an antibiotic-free medium”: "adaptation is mainly driven by the set of resistance mutations present in the T0 strains." Isn't it also possible that mutations are stochastic, and some populations would have higher fitness by chance alone?

9) Figure 3 and subsection “Rapid loss of multi-drug resistance in an antibiotic-free medium”: Are these differences in ratios statistically significant?

10) Subsection “Compensatory mutations rather than reversions dominate”: "identify resistance-conferring SNPs in the T0 population that revert back". What is the likelihood of the exact mutation occurring and invading the population? I would guess this is very low, so searching for the exact mutation would not be a fruitful exercise. How about other mutations in the same genes as those where alterations conferred resistance? The analysis of compensatory mutations reveal that they are located elsewhere in the genome, i.e. not in the resistance genes themselves. This analysis appears superficial. Are these genes somehow functionally related? Part of the same modules? Related by genetic or other types of interactions?

11) Is there any direct evidence that resistance in the clinic comes at a major fitness cost? Also, the role of HGT is not mentioned, although this is a common means of low-cost resistance. Discussion section: The comparison to clinical restriction is problematic for a few reasons. (1) The clinic has an ecological component where the sensitive strain may still persist and invade the resistant strain once antibiotic selection pressure is ceased. (2) Countries are not isolated, so it is unclear that restriction in one place would eliminate pressure on the population as a whole. (3) Many antibiotic adaptations impart cross-resistance, so continued use of other antibiotics may diminish the effect of restricting another.

12) Parallel evolution of the wild type would have provided a useful control to distinguish the change in fitness specifically associated with antibiotic resistance phenotypes from general adaptation to the physicochemical environment of the experiment. Continued propagation in presence of antibiotic would also have provided a useful point of comparison. While these data would have enhanced the manuscript, in their absence, a discussion of expected results from these controls may add depth and perspective to the text.

13) The authors should clarify the caveats and limitations of the study. For example, the results here strongly depend on the fitness cost of resistant mutants acquired from previous evolution in strong selection regime (high antibiotic concentration and large population size) (Lazar Nat Communications 2014). Presumably, the trade-off between resistance level and fitness cost will be different in the low-selection regime and sub-inhibitory concentrations. Thus, the finding of rapid loss of fitness cost may not apply to resistance-conferring mutations evolved under this regime.

---

## [Author Response]

Essential revisions:1) The manuscript is generally well written and referenced satisfyingly. However, the introduction is somewhat lagging behind the rest of the text. Claims in the introduction are not always supported by appropriate citations and quality of the writing could be improved in this section. The authors are referred to some specific comments below for examples of elements of the introduction that could be improved. For instance, the introduction describes an important issue related to the fitness cost of antimicrobial resistance, which is the disconnect between the results of experimental evolution studies and clinical observations. Whether resistance to antimicrobials really does carry a fitness cost, and whether it may or may not be maintained in populations of pathogens thanks to compensatory evolution is indeed controversial, and contradictory evidence currently prevents a definitive answer to this question. This manuscript does not directly address this issue, yet the introduction would suggest otherwise. The authors are invited to revise the introduction to better expose the focus and scope of the study.

Thank you for the suggestion. We modified the introduction with the aim to better describe the reasons for the controversy surrounding implications of clinical and laboratory studies on cost of resistance and the efficacy of compensatory evolution. We also put effort to cite more relevant papers on the subject.

2) Results on the loss of resistance appear overly qualitative, at least in the format used in Figure 3 and in the text (subsection “Rapid loss of multi-drug resistance in an antibiotic-free medium”). We believe a synthetic quantitative analysis may be applied to the results provided in Supplementary file 3 and presented in a visually compelling manner. A quantitative measure of loss in resistance and its causal link or not with loss of cost would be more convincing, and would help readers grasp the magnitude of this phenotypic change. The authors already have MIC data: it is just a matter of using it in a transparent data visualization format, which may still use the qualitative categories, but perhaps as an added layer on top of the quantitative data, rather than as the sole reported measure.

We fully agree and thank you for the suggestion. We designed a new figure depicting resistance level of each T0 and the corresponding T60 strains (Figure 3A).

3) As presented, the results suggest association between fitness recovery and resistance loss. The manuscript would be enhanced by a more formal statistical analysis of this association.

Thank you for the suggestions. As suggested, we also tested the association of relative fitness and the change of resistance against the antibiotic the T0 lines were originally adapted to. Accordingly, we generated a new figure depicting the correlation between fitness recovery and resistance loss. We found a significant negative correlation between fitness improvement and resistance level (Figure 4A).

4) The Discussion section provides a list of evidence that the authors claim supports a driving role for compensatory evolution in adaptation to the antibiotic-free environment. We think that 2 out of the 3 evidences mentioned does not support that claim. The first reads: "First, our analysis has revealed major differences in relative fitness across strains adapted to different antibiotics." We fail to see how differences in fitness between strains supports any role for compensatory evolution. It rather suggests a diversity of molecular mechanisms, which may or may not involve compensation. In other words, this phenomenon appears unrelated to compensation. The next claim which we believe is not supported by the stated evidence is the following: "Third, an in-depth genetic analysis has demonstrated that mutations in marR and envZ increased fitness in the respective antibiotic-resistant strain, but reduced fitness in the wild-type". This is evidence for epistasis, but not for compensation per se.

We modified the text as follows:

“Fitness may increase during the course of laboratory evolution as a result of general adaptation to the environment and/or accumulation of compensatory mutations that mitigate the deleterious side effects of resistance. The second option is more realistic, as several mutations that had accumulated during laboratory evolution affected genes involved in bacterial defense mechanisms against antibiotics or in general stress-responses (e.g. rpoS promoter region/nlpD gene (Stoebel et al., 2009), potD (Shah et al., 2011), soxSR (Jain and Saini, 2016) (Supplementary file 4). An in-depth genetic analysis has also demonstrated epistatic interactions between resistance and putative compensatory mutations in marR and envZ. These mutations had accumulated during laboratory evolution and increased fitness in the respective antibiotic-resistant strain, but reduced fitness in the wild-type (Figure 5A and Figure 5 —figure supplement 1A).”

5) In comparison to the extensive analysis done in Lazar Nat Communications 2014, I would have preferred a similar detailed analysis of the evolutionary experiments in this follow-up work. The broad conclusions are expected, but there are insights which this work could uniquely address. For one, it is interesting to know how the fitness cost and resistance level vary with generation-while there is a broad phenotypic reversion of fitness cost, does this happen early on the evolution, does this reversion occur at the same time that resistance is reacquired (or not)?

Thank you for the suggestion. Due to a strict deadline on submission of the revised manuscript, we focused on four randomly selected lineages only (CIP5a, DOX3a, TMP9e and TOB9e) and measured resistance levels and fitness at three timepoints: the beginning of, the midpoint and the end of the course of laboratory evolution. We found that at certain time points, decline in resistance and fitness recovery are coupled to each other.

Additionally, what is the reproducibility/repeatability of these time courses across the replicates?

We compared the fitness improvements between T60 evolved lines founded from the same T0 genotype versus those founded from different genotypes. As expected, the coefficient of variation was significantly lower (Mann-Whitney *U*-test, *p* < 0.01) for T60 lines founded from the same T0 genotype (median CV = 6.95%) compared to that of founded from different T0 genotypes (median CV = 11.6%).

6) I am curious to know when the mutations found at the end of the experiment (listed in Supplementary file 4) occurred during the lab evolution. Moreover, do the occurrence of specific mutations correlate with the time-changes of fitness cost and resistance level throughout the evolution? This is particularly crucial because there are multiple mutations at the end of the lab evolution and correlating these with fitness cost and resistance level could determine causality and epistasis between mutations.

The issues raised here are indeed very interesting but are beyond the scope of the manuscript. Answering them demands population genomic sequencing and subsequent in-depth phenotypic analyses of clones and populations at different timepoints of laboratory evolution.

7) Why do many of the T60 lines have worse fitness after evolution in antibiotic-free medium? One would expect the fitness to at least stay the same or improve.

This is not entirely unexpected, as the calculated bacterial growth parameters are unlikely to capture all aspects of bacterial fitness.

8) Subsection “Laboratory evolution of antibiotic-resistant strains in an antibiotic-free medium”: "adaptation is mainly driven by the set of resistance mutations present in the T0 strains." Isn't it also possible that mutations are stochastic, and some populations would have higher fitness by chance alone?

There are several reasons to believe that adaptive mutations accumulated during the course of evolution. First, the timescale of laboratory evolution was too limited for fixation of neutral mutations driven by stochastic changes in gene frequencies. Second, the accumulated mutations were far from being random: the corresponding genes are mainly involved in bacterial defense mechanisms against antibiotics or in general stress-responses. Finally, reconstruction of putative compensatory mutations increased fitness in a genotype specific manner, see main text Figure 5A.

9) Figure 3 and subsection “Rapid loss of multi-drug resistance in an antibiotic-free medium”: Are these differences in ratios statistically significant?

Yes, they are. We inserted this result into the corresponding figure legend (Figure 3B):

“Figure 3C and 3D show the number of T60 lines with resistance maintained/lost to the range of antibiotics tested, as absolute number versus ratio, respectively. Doxycycline (DOX) and tetracycline (TET) resistance was frequently lost (Fisher’s Exact test: odds ratio = 0.05, p < 0.001 for DOX and odds ratio = 0.20, p < 0.01 for TET), while aminoglycoside (kanamycin – KAN, tetracycline – TET) resistance was generally maintained in the T60 lines (Fisher’s Exact test: odds ratio = 14.29, p < 0.001 for KAN, odds ratio = 6.5, p < 0.01 for TOB).”

10) Subsection “Compensatory mutations rather than reversions dominate”: "identify resistance-conferring SNPs in the T0 population that revert back". What is the likelihood of the exact mutation occurring and invading the population? I would guess this is very low, so searching for the exact mutation would not be a fruitful exercise. How about other mutations in the same genes as those where alterations conferred resistance? The analysis of compensatory mutations reveal that they are located elsewhere in the genome, i.e. not in the resistance genes themselves. This analysis appears superficial. Are these genes somehow functionally related? Part of the same modules? Related by genetic or other types of interactions?

Due to the low number of mutations that have accumulated during the course of laboratory evolution, we could not perform a rigorous statistical analysis to test this issue. Nevertheless, we provide several examples on functional relatedness between resistance genes mutated in T0 and genes mutated during lab evolution (i.e. found in T60 strains only, Figure 5—figure supplement 1.).

ompR-EnvZ (FOX8b): A cefoxitin-resistant T0 line carries a resistance-conferring mutation in the transcriptional regulatory protein OmpR. OmpR modulates the expression of major outer membrane protein genes, and forms a two-component regulatory system with the sensory histidine kinase EnvZ, a protein mutated in the corresponding T60 line.

PhoQ/SoxR – PhoP/SoxS (TMP9c): A trimetoprim-resistant T0 lines carries a mutation in the sensory histidine kinase PhoQ and the redox‑sensitive transcriptional activator SoxR, whereas the T60 line carried compensatory mutations in genes *phoP* (response regulator in two‑component regulatory system with PhoQ) and *soxS* (superoxide response regulon transcriptional activator).

AcrR-MarR (CIP5a): Mutation in the transcriptional repressor AcrR most likely confers ciprofloxacin-resistance in T0 through activating the AcrA/AcrB/TolC multidrug-efflux system, while a compensatory mutation appeared in MarR, that controls the activity of the *mar* regulon. The *mar* regulon is responsible for regulatingseveral genes involved in antibiotic-resistance, including those that encode components of the AcrA/AcrB/TolC multidrug-efflux system.

11) Is there any direct evidence that resistance in the clinic comes at a major fitness cost?

Yes, there is some prior data supporting these claims, and we cite the relevant publications in the main text, such as: Prabh et al., 2018 and Alzigat et al., 2017

Also, the role of HGT is not mentioned, although this is a common means of low-cost resistance.

We briefly mention HGT in the Discussion section:

“Finally, our study has focused on chromosomal resistance mutations. It is still unclear whether resistance-conferring plasmids are generally lost or the associated fitness costs are mitigated by genomic mutations (Dahlberg and Chao, 2003).”

Discussion section: The comparison to clinical restriction is problematic for a few reasons. (1) The clinic has an ecological component where the sensitive strain may still persist and invade the resistant strain once antibiotic selection pressure is ceased. (2) Countries are not isolated, so it is unclear that restriction in one place would eliminate pressure on the population as a whole. (3) Many antibiotic adaptations impart cross-resistance, so continued use of other antibiotics may diminish the effect of restricting another.

We readily admit these caveats. As regards (1), we write in the Introduction:

“This disagreement between clinical observations and laboratory studies could have multiple reasons. First, antibiotic treatments frequently fail to completely eradicate antibiotic-sensitive bacteria from the population. Following treatment, antibiotic sensitive bacteria with high fitness could rapidly spread in the population, leading to rapid loss of resistance.”

As regards 2) and 3), we write in the Discussion section:

“These considerations must be taken with some caution, as comparison of clinical and laboratory data is not straightforward. For instance, restricted usage of certain antibiotics in hospitals cannot completely eliminate antibiotic selection in a given region due to lack of isolation and cross-resistance between antibiotics.”

12) Parallel evolution of the wild type would have provided a useful control to distinguish the change in fitness specifically associated with antibiotic resistance phenotypes from general adaptation to the physicochemical environment of the experiment. Continued propagation in presence of antibiotic would also have provided a useful point of comparison. While these data would have enhanced the manuscript, in their absence, a discussion of expected results from these controls may add depth and perspective to the text.

As noted above, we addressed this issue as follows:

“Fitness may increase during the course of laboratory evolution as a result of general adaptation to the environment and/or accumulation of compensatory mutations that mitigate the deleterious side effects of resistance. The second option is more realistic, as several mutations that had accumulated during laboratory evolution affect genes involved in antibiotic or general stress response (e.g. rpoS promoter region/nlpD gene (Stoebel et al., 2009), potD (Shah et al., 2011), soxSR (Jain and Saini, 2016) (Supplementary file 4). An in-depth genetic analysis has also demonstrated epistatic interactions between resistance- and putative compensatory mutations in marR and envZ. These mutations had accumulated during laboratory evolution and increased fitness in the respective antibiotic-resistant strain, but reduced fitness in the wild-type (Figure 5A and Figure 5—figure supplement 2A).”

13) The authors should clarify the caveats and limitations of the study. For example, the results here strongly depend on the fitness cost of resistant mutants acquired from previous evolution in strong selection regime (high antibiotic concentration and large population size) (Lazar Nat Communications 2014). Presumably, the trade-off between resistance level and fitness cost will be different in the low-selection regime and sub-inhibitory concentrations. Thus, the finding of rapid loss of fitness cost may not apply to resistance-conferring mutations evolved under this regime.

We describe the caveats of our work in more detail, as follows:

“We should also emphasize, however, that all our evolutionary experiments were performed in devoid of any antibiotics. The next logical step is to study the occurrence of phenotypic reversion during exposure to temporarily changing antibiotic treatments or sublethal antibiotic dosages as it may occur in real-life situations (Andersson and Hughes, 2014). Our study leaves open the question about the extent by which initial fitness costs of resistance mutations and/or intensity of antibiotic selection shapes subsequent compensatory evolution and associated loss of resistance. Finally, our study has focused on chromosomal resistance mutations. It is still unclear whether resistance-conferring plasmids are generally lost or the associated fitness costs are mitigated by genomic mutations (Dahlberg and Chao, 2003).